# The First Exposure Assessment of Mercury Levels in Hair among Pregnant Women and Its Effects on Birth Weight and Length in Semarang, Central Java, Indonesia

**DOI:** 10.3390/ijerph191710684

**Published:** 2022-08-27

**Authors:** Muflihatul Muniroh, Saekhol Bakri, Ainun Rahmasari Gumay, Julian Dewantiningrum, Mulyono Mulyono, Hardian Hardian, Megumi Yamamoto, Chihaya Koriyama

**Affiliations:** 1Department of Physiology, Faculty of Medicine, Diponegoro University, Semarang 50275, Indonesia; 2Department of Public Health, Faculty of Medicine, Diponegoro University, Semarang 50275, Indonesia; 3Department of Epidemiology and Preventive Medicine, Graduate School of Medical and Dental Sciences, Kagoshima University, Kagoshima 890-8520, Japan; 4Department of Obstetrics and Gynecology, Faculty of Medicine, Diponegoro University, Semarang 50275, Indonesia; 5Department of Pediatrics, Faculty of Medicine, Diponegoro University, Semarang 50275, Indonesia; 6Department of Environment and Public Health, National Institute for Minamata Disease, Minamata, Kumamoto 867-0008, Japan

**Keywords:** hair mercury level, pregnant women, birth weight and length, Indonesia, fish consumption

## Abstract

(1) Background: Methylmercury (MeHg) exposure during pregnancy is an important issue due to its possible adverse health effects on fetus. To contribute the development of assessment system of Hg exposure through fish consumption and health effects on children, we examined the hair Hg levels in pregnant women and birth weight and length. (2) Methods: In 2018, a cohort study was conducted on 118 pregnant women-infant pairs from six community health centers in the northern coastal area in Central Java Indonesia. Data on mothers’ characteristics during pregnancy, birth outcomes, and fish consumption were collected. Total Hg concentrations were determined from hair samples. (3) Results: The median (min-max) of the maternal hair Hg level was 0.434 (0.146–8.105) µg/g. Pregnant women living in lowland areas, near the sea, showed higher hair Hg concentration and fish consumption than those in highland areas {[0.465 (0.146–8.105) vs. 0.385 (0.150–1.956) µg/g; *p* = 0.043] and [(85.71 (0–500.0) vs. 49.76 (0.0–428.57) g/day; *p* < 0.01], respectively}. The maternal hair Hg level had no association with baby’s birth weight and length. (4) Conclusions: The median maternal hair Hg is at a low level and had no association with infant birth weight and length in this study subjects.

## 1. Introduction

Mercury (Hg) is widely distributed in the environment, and humans can be exposed to it through multiple pathways such as inhalation and ingestion, in multiple exposure forms including elemental (Hg^0^), inorganic (Hg^2+^), and organic [methylmercury (CH_3_Hg^+^: MeHg), etc.] [1,2,3]. All forms of Hg have in common a toxic effect, particularly MeHg on the growth and neurodevelopment of the fetus and child. MeHg is a well-known neurotoxicant that primarily targets the brain because it is easily absorbed in the digestive tract through the consumption of seafood and migrates to the central nervous system after passing through the blood–brain barrier and blood-placental barrier; therefore, MeHg exposure to the child begins during pregnancy and it potentially impacts fetal development [4,5,6]. Thus, it is an important issue in the public health field to assess the exposure to MeHg and evaluate the associated health effects on the population.

An important source of MeHg exposure especially during pregnancy is the con-sumption of fish and marine mammals. To prevent pregnant women from being exposed to high Hg levels, the Japanese Government, the U.S. Food and Drug Administration (FDA), and the Environmental Protection Agency (EPA) has issued recommendations on the types of fish and their intake to avoid during pregnancy because of higher Hg levels, such as bottlenose dolphin, king mackerel, marlin, orange roughy, shark, swordfish, tilefish, and tuna bigeye. [7,8]. The EU-Scientific Committee for Food (EU-SCF) and Joint FAO/WHO Expert Committee on Food Additives (JFCFA) have recommended the tolerable weekly intake of Hg as 0.3 mg [9]. The Ministry of Health, Labor, and Welfare (MHLW), Japan has set an upper limit of 2.8 µg/g for the hair Hg level as the tolerable in-take for mothers considering the effects on the fetus [7]. The U.S. EPA, on the other hand, has a more stringent standard level and recommends hair Hg levels of 1.1 µg/g or less for pregnant women [10]. 

Exposure to MeHg at high concentrations has been associated with neurodevelop-mental problems in the fetus; however, its effects at low concentrations are not well un-derstood yet [11,12,13,14]. Some studies reported a possibility of adverse effects on fetal development even at low levels of MeHg exposure [15]. Kobayashi et al. also reported that prenatal exposures to low Hg levels have been correlated with low birth weight and premature birth, which are important factors in under-five mortality and cause developmental problems in children [16]. Previous studies from Minamata and Niigata, Japan, and Iraq reported the severe neurological effects on the children when exposed to high levels (>10 µg/g) of MeHg in utero [1]. Studies in the U.S. and Poland suggested that even low levels (≤1.2 µg/g) of Hg in maternal hair, blood, or cord blood might be associated with devel-opmental delay among infants [17,18]. Previous studies reported the high Hg levels in maternal blood in the first and second trimester [mean (standard deviation): 6.06 (3.81) and 4.99 (3.45) μg/L, respectively] were negatively correlated with birth weight, and a doubling effect of total Hg from cord blood (8.2 μg/L) was associated with a 7.7 g reducetion of placental weight, 0.052 cm of head circumference, 14.1 g of birth weight, and 0.047 cm of length birth, which indicates the possible effect of MeHg exposure during pregnancy on fetal small birth size and growth restriction [5,16,19]. In particular, exposure to MeHg may cause a reduction in birth weight and head circumference, which may increase the risk of brain developmental disorders [19]. On the other hand, another study reported that exposure to MeHg (5.9 µg/g, maternal hair, which is equivalent to 17.15 μg/L in the blood) did not affect birth weight [20]. Thus, there is still no clear conclusion as to whether low concentrations of MeHg have health effects on the fetus. Therefore, the assessment of prenatal MeHg exposure in various populations is an important issue to be investigated [3,15]. 

The clarification of the balance of risk of MeHg exposure and nutritional benefit including fatty acids such as docosahexaenoic acid from maternal prenatal fish consumption is important for child cognitive development [21]. Since dietary habits including fish vary from country to country and region to region, it is important to determine the recommended levels of seafood intake for each country (population) concerning seafood intake in pregnant women [22,23]. Although fish consumption in Indonesia has increased [24], no studies reported the assessment of MeHg exposure to pregnant women through the consumption of fish and shellfish.

Semarang is the capital city of Central Java, Indonesia, consisting of highland, mountainous areas, and lowlands near the coastline. Differences in residential areas may also affect the amount of fish consumed by the people living there. This is the first study to evaluate prenatal Hg exposure among pregnant women and its association with birth weight and length in Indonesia. Therefore, the present data are important in assessing whether Hg exposure is a situation of concern for maternal and child health care measures in the Indonesian population. 

In the current study, we evaluated the prenatal MeHg exposures of pregnant women through fish consumption, and its effect on birth weight and length in Semarang, Central Java Indonesia. The result of this study will contribute to the development of an assessment system for MeHg exposure among pregnant women, particularly in Indonesia.

## 2. Materials and Methods

### 2.1. Research Subjects

This cohort study was conducted in 118 pregnant women-infant pairs from six different community health centers located in the northern coastal area in Semarang, Central Java Indonesia, in 2018. The inclusion criteria for the subjects in this study were pregnant women in the 2nd trimester with a singleton pregnancy, aged less than 40 years old, and living in their current home throughout their pregnancy. After being given a detailed explanation about this study, written informed consent was obtained from each participant. All participants underwent the anamnesis and physical examination to obtain the data on age, body mass index (BMI), gestational age, upper arm circumference, blood pressure, living area (highland: altitude at 90–359 m above sea level, lowland: altitude at 0.75–3.5 m above sea level) [25], income, and mothers’ education. Newborns’ birth weight and length were measured 5 min after delivery. At first, 130 pregnant women participated in this study, but 11 dropped out due to moving to other cities (*n* = 7), loss of contact (*n* = 3), and having miss-carriage (*n* = 1). One of the participants turned out to be 14 years old, and we excluded this woman because of the WHO criteria of women in reproductive age (15–49 years) [26]. Therefore, the final number of subjects was 118 pairs of mothers and newborns. 

### 2.2. Measurement of Hair Total Mercury (T-Hg)

Except for cases such as adhesion of inorganic mercury, the chemical form of more than 90% of total mercury (T-Hg) in hair is MeHg; therefore, the level of MeHg exposure from the fish consumption, which is the main source of Hg exposure for the general population, can be assessed by measuring total mercury in hair [1,27].

Scalp hair samples were collected from all participants by cutting about 1 cm from the scalp to obtain the information of Hg exposure in early pregnancy, where 1 cm represents 1-month exposure [28]. The hair samples were stored in a labeled transparent plastic clip until the Hg measurement. T-Hg was determined using a reducing-vaporization mercury analyzer HG-1500 (Sanso Co., Tokyo, Japan) at Kagoshima University, Kagoshima, Japan. The preparation procedures followed the previous studies [29,30]. Before running the Hg measurement, hair samples were washed with acetone 5 mL for 30 min using clean bottle, and then dried and kept in the desiccator. The cleaned hair samples were digested (approximately 10 mg, not more than 5 cm in length), and a 0.4 mL mixed solution of nitric acid and perchloric acid (1:1 ratio) and 1 mL sulfuric acid were added. The specimens were heated at 200 °C for 30 min and cooled under running water. The volume was adjusted to 10 mL using distilled water. The total Hg level was then measured using the cold vapor atomic absorption (CVAA) method, and a 10% SnCl_2_ solution was added. Before the T-Hg analysis of each hair sample, certified reference material (CRM) measurements were performed to confirm that the analytical system was running properly. Duplicate reagent blanks and Hg standard solutions were used in each analysis of hair samples. The measurement was repeated five times. The CRM of hair (NIES CRM No.13) was measured as quality control (since the hair Hg levels of the subjects in this study had a wide range, such as 0.146–8.105 μg/g), and the determined T-Hg level of 4.369 ± 0.049 µg/g was within the certified range of 4.42 ± 0.20 µg/g. A coefficient of variation (CV) of 1.13% was obtained from five times repeated measurements using the CRM.

### 2.3. Estimation of Fish Consumption

Fish consumption was estimated using the answers to the questionnaire. Participants were asked about their types and frequencies of daily fish and other seafood intakes during the pregnancy by using pictures of fish and seafood in the questionnaire and about the quantities by using food models of fish (50 g) [29]. Ten types of fresh-water, eight marine fish, and six other marine species that are commonly consumed in Semarang were included in the questionnaire. We summed up the total fish intake per week, and then divided by 7 to obtain the daily amount of fish intake. The daily amount of fish consumption was categorized into quartiles of <40, 40–79, 80–159, and >160 g/day.

### 2.4. Statistical Analysis

The obtained data for Hg concentration were analyzed with characteristic data of the mother and newborn babies to determine the correlation between the Hg level with pregnancy conditions and babies’ outcomes. The normality data were examined using the Kolmogorov–Smirnov test. The quantitative data were presented as median and min-max (minimum-maximum), and the qualitative data were shown as number and percent. The comparison between low and highland related to maternal characteristics and baby’s outcome was analyzed using Fisher exact test and Chi-square test. Even after log-transformation was applied for Hg data, the data were still not normally distributed, then the difference in log10 Hg levels between groups was examined using the Mann–Whitney U test and Kruskal–Wallis tests, as the data were non-parametric. The relationship between maternal hair Hg levels and maternal characteristics and babies’ outcomes were analyzed using univariate regression analysis. Multiple linear regression analysis was used to examine the relationship between prenatal mercury levels and offspring birth weight and length, controlling for possible effect modifiers, such as newborn sex, gestational age, BMI before pregnancy, and education (the category of primary/junior high school was used as reference). The covariables for the multiple regression model were selected from univariate analysis result of the baby weight. Gestational age, BMI before pregnancy, and education were significantly associated with weight babies. The sex variable was added because of its influence on birth weight and length. Then, they were also applied for the covariables in the baby’s length calculation. All statistical tests were conducted using R, and the significance level was <0.05.

### 2.5. Ethical Aspects

The study has been carried out by the Code of Ethics of the World Medical Association (Declaration of Helsinki). Informed consent was obtained from all respondents as a prior condition of inclusion as subjects of the study. Ethical clearance was approved by the Health Research Ethics Committee, Faculty of Medicine Universitas Diponegoro with the study reference number 391/EC/FK-RSDK/V/2018 and Kagoshima University Graduate School of Medical and Dental Sciences.

## 3. Results

### 3.1. Characteristics of the Study Subjects and Area

The study subjects were 118 pregnant mother-infant pairs who were living in Semarang, Central Java area during their pregnancy (Table 1). Around 60% (*n* = 70) of the mother were living in lowland areas and 40% (*n* = 48) in highland areas. The median age of the mothers was 29.5 years old, and women living in the lowlands (30.5 years old) were slightly older on average than those living in the highlands (28 years old). More than half of the study subjects were within a normal range of BMI (55.1%) and mid-upper arm circumference (85.6%). All subjects did not have a history of smoking and alcohol drinking. Although 36 women were null parity, two of them had a history of abortion. Thus, the number of first pregnancies in this study was 34. Approximately 20% (*n* = 17) of subjects had several complications in their previous pregnancies, such as preeclampsia, abortion, intrauterine fetal death (IUFD), intrauterine growth restriction (IUGR), premature rupture of the membrane, placenta previa, premature and Gemelli, macrosomia, and obstructed labor, and these were found at a higher frequency in lowland women (n = 14) than in highland (*n* = 3) (*p* = 0.154). Women in lowland (25 Indonesian Rupiah (IDR) · 10^−5^/month) had a significantly higher income than in highland areas (23 IDR · 10^−5^/month) (*p* = 0.002). The number of university education level and working mother in lowland (n = 10 and 41, respectively) were higher than in highland (n = 7 and 28, respectively), but it was not statistically significant.

Figure 1 shows the distribution of the subject living areas in Semarang, Central Java, Indonesia. Seventy pregnant women (59.3%) were recruited at four primary health centers in the lowland area, and the rest were from two primary health centers in the highland.

Table 2 shows the distribution of birth weight and length in this study. The median birth weight and length of babies were 3100 g and 49 cm, respectively. There was no difference in birth weight and length between lowland and highland areas. Fifteen babies (12.7%) were preterm delivery, where labor was before 37 weeks of gestation. Fifty-one babies (43.2%) were delivered by Cesarean section. Mothers who had caesarean deliveries had a significantly higher BMI before pregnancy (*n* = 19) (median = 23.9, min-max =14.7–41.1) than those who had vaginal deliveries (*n* = 16) (median = 21.9, min-max = 14.4–38.1) (*p* = 0.035, Mann Whitney U test; data was not shown). Women with a history of severe complications in previous pregnancies tended to show a higher proportion of cesarean section (11/16, 68.8%) than those without a history of complications (30/69, 43.5%) (*p* = 0.276, Fisher’s exact test; data were not shown). 

### 3.2. Comparison of Hg Concentrations in Mother Conditions

As described in Table 3, the median level of hair Hg in pregnant women was 0.434 (0.146–8.105) µg/g, nine subjects (8%) showed more than 1.1 µg/g, and three subjects (2.5%) more than 2.8 µg/g of Hg concentration ranging from 1.279 to 8.105. Mothers living in lowland areas had higher hair Hg concentrations than those living in highland areas (*p* = 0.043 by Mann–Whitney U test). Other factors were not related to hair Hg concentration. The median values of fish, both marine and freshwater fish, and seafood, except fish, intake in the study subjects were 77.1 g/day (539.7 g/week) and 28.6 g/day (200.2 g/week), respectively, and none of them were correlated with hair Hg concentration (Spearman-rho: 0.090 and 0.165, respectively). The women with the highest level of hair Hg (8.105 µg/g) had a history of hair treatment and did not consume fish during the pregnancy.

Figure 2 shows the distribution of a Log10 Hg level and fish consumption in pregnant mothers, and Figure 3 shows a significantly high fish intake in women living in lowland areas compared with those in highland areas; Median (min-max): 85.71 (0–500) and 49.76 (0–428.57) g/day, respectively (*p* = 0.002 by Mann–Whitney U test).

### 3.3. Association of Maternal Hair Hg Concentrations with Birth Weight and Length of Babies

Table 4 shows the results of univariate analyses for the association between maternal hair Hg levels with birth outcomes, namely birth weight and length. Mothers’ body weight at the survey and before the pregnancy was positively related to both birth length and weight. Furthermore, mothers’ BMI before the pregnancy, education, and gestational age were significantly related to birth weight. 

Multivariate regression analyses (Table 5) showed a positive trend between maternal hair Hg and birth weight (β coefficient = 154.6; 95% CI = −73.29–382.5; standard error = 115.0; *p* = 0.182), and an opposite effect on birth length (β coefficient = −0.089; 95% CI = −1.518–1.340; standard error = 0.721; *p* = 0.902), after adjusting the effects of babies’ gender, gestational age, and maternal BMI before pregnancy and education, but the association was not statistically significant.

## 4. Discussion

As far as we know, this is the first report of MeHg exposure during a prenatal period among pregnant women and the effects on birth weight and length of their babies in Indonesia. Regarding the potential effects of MeHg prenatal exposure in the coastal population, this study examined hair Hg levels from the maternal community either living far from or near the coastal area of Java Sea in Semarang Central Java Indonesia. The median (min-max) of hair Hg level of 118 pregnant women in this study was 0.434 (0.146–8.105) µg/g, which is within the reference dose of the US EPA (less than 1.1 µg/g hair) and that of the MHWL, Japan (2.8 µg/g) [10]. Although nine subjects had Hg concentrations in the hair above 1.1 µg/g, and three subjects above 2.8 µg/g, ranging from 1.279 to 8.105, there was no effect on pregnancy such as a miscarriage, and her baby showed a normal APGAR Score (9–10) and within a normal range of birth weight (≥2500 g) and length (≥50 cm). A 22-year-old woman with her first pregnancy had a miscarriage at 22 weeks of gestational age, and her hair Hg level was 0.589 µg/g. Previous studies reported an association between Hg exposure and miscarriage, even below the toxic dose limits [34,35], but since the number of miscarriages in our study was only two, we could not claim that there was no association between Hg exposure and miscarriage. The further study should be done in a greater number of cases. The highest hair Hg level, 8.105 µg/g, was observed. Although this subject lived in a lowland area, she was a housewife and had no history to work for the last five years, and had not consumed any fish at all during her pregnancy. Therefore, we cannot deny the possibility that her high Hg level might be due to another reason such as external factors. One of the possibilities is hair treatment containing Hg; however, we could not obtain more details such as the product’s name and ingredients. To exclude the effect of an external contamination, the hair samples were washed with acetone before Hg measurement. Even after excluding this outlier, there was a marginal difference in hair Hg levels among women between low and highland areas (*p* = 0.055). 

A positive, though not significant, association between maternal hair Hg concentration and birth weight, as shown in this study, suggests that maternal Hg level did not have the effect of reducing the birth weight at this exposure level (median = 0.434 µg/g). This is consistent with the results in previous studies [16,36,37]. Previous studies on Hg effects on fetal growth have been inconsistent. Prenatal exposure to MeHg (mean hair Hg levels 3.92 µg/g) in Seychelles was not related to birth weight [20]. 

An infant’s birth weight is an indicator of maternal nutrition during pregnancy [38]. The baby’s birth weight is also influenced by other factors, including the exposure to tobacco smoke and alcohol drinking during pregnancy, education, and socioeconomic status [39,40]. In this study, however, most of the subjects did not have these risk factors, and only maternal BMI was significantly related to infants’ birth weight. The income factor, as an important factor that can influence the maternal nutrition intake during pregnancy, among lowland and highland women, was also analyzed in a multiple regression model, and showed that it was not associated with birth weight and length. The birth weight is also related to delivery time, and, in this study, 12.7% of mothers had pre-term delivery. 

The prevalence of Cesarean section in this study (43.2%) was higher than that in another report of Indonesia (17.25%) and WHO recommended range (10–15%) [41,42]. The main indications in these cases were mostly related to medical issues, such as dystocia, severe pre-eclampsia, and cardiovascular problems. Since the study area is the capital city, it has a proper referral system to appropriate medical facilities, which is also responsible for the relatively high number of cesarean sections among the subjects. Although the study subjects were collected from primary health centers, we cannot deny the possibility that mothers who were particularly concerned about their health have visited there, and, as a result, there may have been many cases of cesarean sections. 

The US EPA recommends that the upper limit of Hg levels in cord blood should be below 5.8 µg/L [43]. However, because the concentration of MeHg in fetal blood is approximately 1.7 times higher than that in maternal blood, some studies consider 3.5 µg/L as the safe upper-limit value for the maternal blood MeHg level [44,45]. The ratio of the Hg concentration in the brain to that in the blood has been reported that corresponds to 5 to 1, and the ratio of Hg concentration in the hair to that in the blood is approximately 1 in 250 [11,46,47], indicating that 0.875 µg/g of Hg in hair is equivalent to 3.5 µg/L of Hg in blood. In this study, we found that hair Hg levels among Indonesian pregnant women were low, 0.434 µg/g, and it is similar to those reported in the US and Poland [17,18,48]. It is noteworthy, however, that these previous studies found significant positive associations with neurocognitive impairments among children around 6 months old–one-year-old. 

In this study, women living in lowland areas tended to eat more fish, and their hair Hg level was higher than that among women in highland areas. However, the hair Hg level was not significantly correlated with the consumption of fish or seafood (Spearman rho = 0.090 and 0.165, respectively). We also checked the data using the different category of frequency (<1×, 1–3×, >3× per week), but the conclusion was not changed. Therefore, we used only daily fish consumption. The hair Hg levels in the current study subjects were relatively low although the median value of fish intake in this study was 73.1 g/day (511.7 g/week), which is above the recommended fish amount for pregnant women by the U.S. FDA, 224 to 336 g (8–12 oz in 2–3 serving), and the MHWL Japan, 10 to 160 g (15–80 g in 1–2 serving) per week [7,23]. In this study of subjects, the most frequently consumed types of fish are milkfish, pomfret fish, and mullet fish. Milkfish and mullet fish are marine fish, and pomfret fish is a freshwater fish. According to the results of this study, Hg levels in fish in the study area may be relatively low. One of the limitations of this study is that we were not able to obtain the specific information on the consumption of each fish type because of the mothers not being able to remember the detail information about that, and freshwater fish was considered not to contain a higher Hg level. Another limitation is a lack of information on Hg levels in fish. A detailed study of the fish species eaten by the residents in this area may help to clarify the cause of this discrepancy.

To our knowledge, there is no study reporting Hg levels in these fish caught in Indonesia, particularly in the study area. Seafood contains high-quality protein and highly unsaturated fatty acids such as eicosapentaenoic acid and docosahexaenoic acid [49]. It is also a good source of micronutrients such as calcium and thus has excellent nutritional properties [50]. However, it is also true that some fish contain high Hg levels, which pregnant women should be cautious about [10,51]. Therefore, this study will contribute to the establishment of an assessment system for MeHg exposure among pregnant women and risk assessment in a possible high-level of MeHg exposed populations such as the high number of fish and whale-eating areas in Indonesia.

## 5. Conclusions

In this study, we have reported for the first time the assessment for MeHg exposure that is likely due to fish consumption in pregnant women and its effects on birth outcomes in Indonesia. There was no correlation between maternal hair mercury levels at the median (min-max) of 0.434 (0.146 to 8.105) µg/g.

## Figures and Tables

**Figure 1 ijerph-19-10684-f001:**
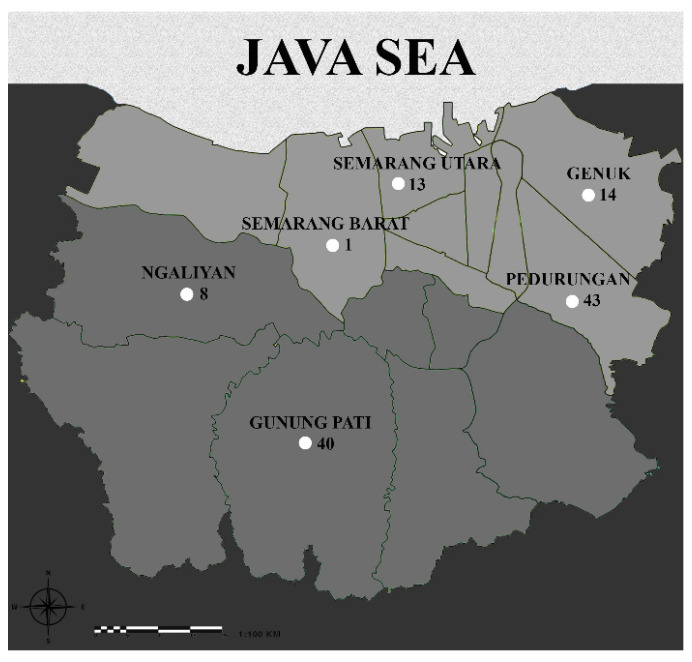
The distribution of study area and subjects in Semarang, the capital city of Central Java Indonesia; lowland (0.75–3.5 m below sea level; dark grey color) and highland (90–359 m above the sea level; light grey color), sea (white color). The subjects were 70 from lowland areas; Semarang Utara (13), Semarang Barat (1) Genuk (14), and Pedurungan (43), and 48 from highland areas; Ngaliyan (8) and Gunung Pati (40).

**Figure 2 ijerph-19-10684-f002:**
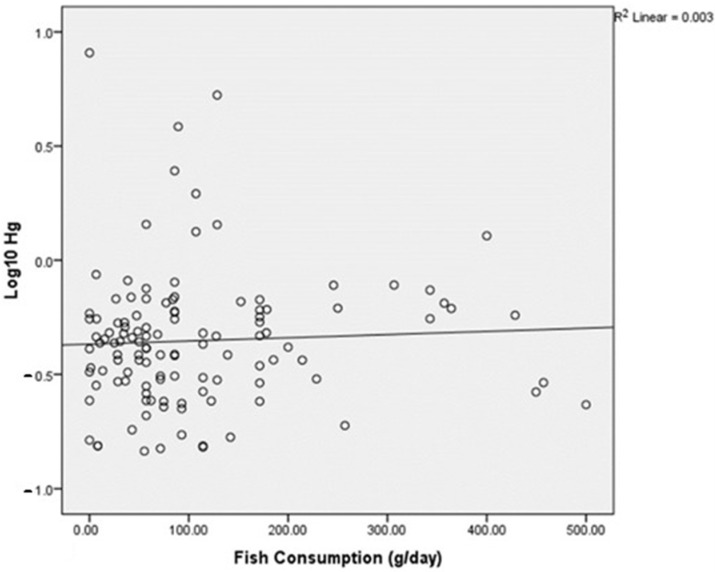
The scatter plot of Log10 Hg and fish consumption.

**Figure 3 ijerph-19-10684-f003:**
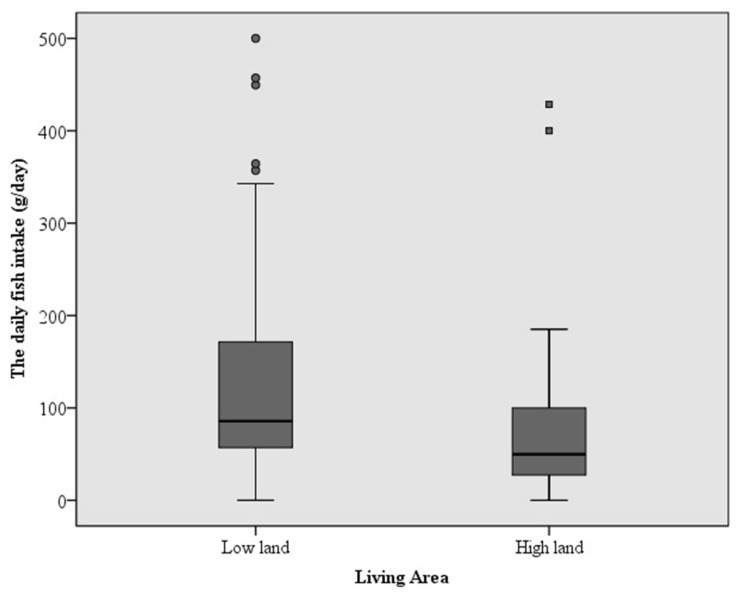
Comparison of daily fish consumption between subjects living in lowland and highland areas. The *p*-value was 0.002 by the Mann–Whitney U test.

**Table 1 ijerph-19-10684-t001:** Maternal characteristics of the study population.

	Median (Min-Max), Number (%)			*p*-Value
Variable	Total (*n* = 118)	Living Area *		
		Lowland (*n* = 70)	Highland (*n* = 48)	
Age (year)	29.5 (19–39)	30.5 (19–39)	28 (19–35)	0.061 ^a^
<30	59 (50.0)	32 (45.7)	27 (56.3)	0.349 ^b^
≥30	59 (50.0)	38 (54.3)	21 (43.8)	
BMI before pregnancy (kg/m^2^)	22.8 (14.4–41.1)	23.5 (14.4–41.1)	21.8 (15–39.7)	0.153 ^a^
<18.5	18 (15.3)	9 (12.9)	9 (18.8)	0.828 ^b^
18.5–24.9	65 (55.1)	39 (55.7)	26 (54.2)	
25–29.9	25 (21.2)	16 (22.9)	9 (18.8)	
≥30	10 (8.5)	6 (8.6)	4 (8.3)	
Mid upper arm circumference (cm) **	26.5 (20.0–37.1)	26.5 (20–37.1)	26.5 (20–36)	0.624 ^a^
<23	17 (14.4)	9 (12.9)	8 (16.7)	0.755 ^b^
≥23	101 (85.6)	61 (87.1)	40 (83.3)	
Parity before this pregnancy				0.093 ^b^
0	36 (30.5)	18 (25.7)	18 (37.5)	
1	56 (47.5)	32 (45.7)	24 (50)	
≥2	26 (22.0)	20 (28.6)	6 (12.5)	
Presence of disease in previous pregnancy ***				0.154 ^c^
Yes	17 (20.0)	14 (25.5)	3 (10.0)	
No	68 (80.0)	41 (74.6)	27 (90.0)	
Education				0.521 ^b^
Elementary-junior	38 (32.2)	20 (28.6)	18 (37.5)	
High school	70 (59.3)	43 (61.4)	27 (56.3)	
Universities	10 (8.5)	7 (10.0)	3 (6.3)	
Income (IDR.10^−5^ /month) ****	23.5 (5–78)	25 (5–78)	23 (5–50)	0.002 ^a^
<23	43 (36.4)	20 (28.6)	23 (47.9)	0.051 ^b^
≥23	75 (63.6)	50 (71.4)	25 (52.1)	
Occupation				0.524 ^c^
Labourer	5 (4.2)	3 (4.3)	2 (4.2)	
Employees	28 (23.7)	19 (27.1)	9 (18.8)	
Entrepreneur	8 (6.8)	6 (8.6)	2 (4.2)	
Housewife	77 (65.3)	42 (60.0)	35 (72.9)	

* Living area: lowland (0.75–3.5 m below sea level); highland (90–359 m above the sea level) [25], ** A reference normal range for pregnant mother in Asian countries, including Indonesia [31], *** Subjects who have never given birth were excluded. Diseases during the pregnancy were preeclampsia (*n* = 2), abortion (*n* = 2), intrauterine fetal death (*n* = 2), intrauterine growth restriction (*n* = 1), premature rupture of membrane (*n* = 6), placenta previa (*n* = 1), premature and Gemelli (*n* = 1), macrosomia (*n* = 1), and obstructed labor (*n* = 1), **** Minimal standard of monthly income in Semarang, Central Java Indonesia (equal to 158.08 USD) [32], ^a^: Mann–Whitney U test; ^b^: Chi-square test; ^c^: Fisher’s exact test.

**Table 2 ijerph-19-10684-t002:** Newborn characteristics of the study population.

	Median (Min-Max), Number (%)			
Birth Outcomes	Total	Living area		*p*-Value
		Lowland	Highland	
Birth weight * (g)	3100 (2260–4200)	3100 (2260–4200)	3120 (2260–3830)	0.748 ^a^
<2500	5 (4.2)	2 (2.9)	3 (6.3)	0.665 ^b^
≥2500	113 (95.8)	68 (97.1)	45 (93.7)	
Birth length * (cm)	49 (44–59)	49 (45–58)	49 (44–59)	0.315 ^a^
<50	73 (61.9)	46 (65.7)	27 (56.3)	0.397 ^b^
≥50	45 (38.1)	24 (34.3)	21 (43.7)	
Gender				0.937 ^b^
Boy	65 (55.1)	35 (50.0)	30 (62.5)	
Girl	53 (44.9)	35 (50.0)	18 (37.5)	
Gestational age at birth (week)	38.5 (34–42)	39 (34–42)	38 (34–42)	0.937 ^a^
<37	15 (12.7)	9 (12.9)	6 (12.5)	1.000 ^b^
≥37	103 (87.3)	61 (87.1)	42 (87.5)	
Delivery process				0.030 ^b^
Vaginal	67 (56.8)	34 (48.6)	33 (68.8)	
Caesarean	51 (43.2)	36 (51.4)	15 (31.3)	

* Categorized by WHO [33], ^a^: Mann–Whitney U test; ^b^: Chi-square test.

**Table 3 ijerph-19-10684-t003:** Comparison of total hair Hg concentrations concerning several maternal conditions.

Variable	N	Hair Hg Concentration (µg/g)						*p*-Value
		P5	P10	P50	P90	P95	Min-Max	
All subject	118	0.154	0.207	0.434	0.802	1.461	0.146–8.105	
Age (year)								
<30	59	0.152	0.163	0.465	0.814	2.463	0.146–5.290	0.741 ^a^
≥30	59	0.224	0.236	0.409	0.776	1.431	0.154–8.105	
Living area								
Lowland	70	0.200	0.241	0.465	0.813	3.085	0.146–8.105	0.043 ^a^
Highland	48	0.153	0.154	0.385	0.747	1.389	0.150–1.956	
Education								
Elementary–Junior	38	0.180	0.223	0.450	1.431	2.604	0.152–5.290	0.269 ^b^
Senior High school	70	0.154	0.174	0.414	0.689	1.302	0.146–8.105	
Universities	10	0.163	0.175	0.596	1.847	1.956	0.163–1.956	
Income (IDR.10^−5^/month)								
<23	43	0.158	0.192	0.385	0.792	1.414	0.152–3.845	0.297 ^a^
≥23	75	0.154	0.212	0.465	0.834	2.057	0.146–8.105	
Hair treatment								
Yes	14	0.150	0.152	0.400	5.975	8.105	0.150–8.105	0.351 ^a^
No	104	0.169	0.226	0.437	0.790	1.406	0.146–5.290	
Fish consumption (g/day)								
<40	31	0.154	0.179	0.434	0.787	3.761	0.154–8.105	0.314 ^b^
40–79	30	0.148	0.184	0.385	0.687	1.059	0.146–1.435	
80–159	30	0.153	0.168	0.447	2.412	4.495	0.152–5.290	
≥160	27	0.207	0.239	0.535	0.776	1.079	0.189–1.279	
Seafood consumption								
Yes	90	0.164	0.224	0.446	0.860	1.670	0.150–5.290	0.287 ^a^
No	21	0.147	0.157	0.365	0.579	0.735	0.146–0.751	
Not eat fish	7	0.163	0.163	0.409	8.105	8.105	0.163–8.105	

^a^: Mann–Whitney U test; ^b^: Kruskal–Wallis test, P = percentile.

**Table 4 ijerph-19-10684-t004:** Results of univariate regression analysis for birth length and weight of babies.

Variable	Birth Length (cm)			Birth Weight (g)		
	β Coefficient (CI: 95%)	SE	*p*-Value	β Coefficient (CI: 95%)	SE	*p*-Value
Mothers						
Age (year)	0.037 (−0.046–0.119)	0.042	0.383	6.594 (−8.249–21.44)	7.494	0.381
Body weight at the survey (kg)	0.039 (0.006–0.072)	0.017	0.021	10.72 (4.959–16.49)	2.910	<0.001
Body weight before pregnancy (kg)	0.034 (0.001–0.067)	0.017	0.048	10.01 (4.205–15.81)	2.929	0.001
Height (cm)	0.051 (−0.022–0.126)	0.037	0.166	11.21 (−2.017–24.43)	6.677	0.096
BMI before pregnancy (kg/m^2^)	0.067 (−0.018–0.152)	0.043	0.119	22.75 (7.935–37.57)	7.482	0.002
Income (IDR.10^−5^/month)	−0.002 (−0.034–0.031)	0.016	0.908	1.887 (−3.918–7.693)	2.931	0.521
Highland (vs. lowland)	0.347 (−0.492–1.187)	0.424	0.415	−61.26 (−212.2–89.69)	76.22	0.423
Education (vs. senior high school)						
Elementary-junior	−0.165 (−1.073–0.7425)	0.458	0.719	84.92 (−73.23–243.1)	79.84	0.290
Universities	0.271 (−1.252–1.795)	0.769	0.725	370.2 (104.8–635.6)	133.9	0.007
Fish consumption (g/day)	−0.002 (−0.006–0.002)	0.002	0.385	−0.519 (−1.210–0.172)	0.349	0.139
Hair Hg, log-Hg (µg/g)	−0.043 (−1.453–1.366)	0.712	0.952	204.96 (−45.65–455.6)	126.5	0.108
Babies						
Boy (vs. girl)	0.226 (−0.604–1.057)	0.419	0.590	103.3 (−45.03–251.5)	74.87	0.171
Gestational age (week)	0.229 (−0.037–0.495)	0.134	0.090	90.23 (44.75–135.7)	22.96	<0.001
Caesarian (vs. vaginal)	−0.132 (−0.966–0.703)	0.421	0.755	29.58 (−120.4–179.6)	75.74	0.697

CI = confidence interval; SE = standard error.

**Table 5 ijerph-19-10684-t005:** The association between maternal hair mercury levels with birth weight and length of babies.

Variable	β Coefficient	CI 95%	SE	*p*-Value
Birth weight (g)				
Hair mercury, log-Hg (µg/g)	154.6	−73.29–382.5	115.0	0.182
Boy	72.27	−61.74–206.2	67.62	0.288
Gestational age (week)	75.10	31.24–119.0	22.14	0.001
BMI before pregnancy (kg/m^2^)	19.99	6.111–33.87	7.01	0.005
Education				
High school	−76.76	−222.9–69.40	73.75	0.300
Universities	254.5	−37.33–512.7	130.3	0.053
Birth length (cm)				
Hair mercury, log-Hg (µg/g)	−0.089	−1.518–1.340	0.721	0.902
Boy	0.115	−0.725–0.955	0.424	0.786
Gestational age (week)	0.196	−0.079–0.471	0.139	0.161
BMI before pregnancy (kg/m^2^)	0.062	−0.026–0.149	0.044	0.164
Education				
High school	0.168	−0.749–1.084	0.463	0.717
Universities	0.404	−1.216–2.023	0.817	0.622

Adjusted by gender, gestational age, BMI before pregnancy, and education (the category of primary/junior high school was used as reference); CI = confidence interval; SE = standard error.

## Data Availability

Not applicable.

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
