# Peer review of "The First Exposure Assessment of Mercury Levels in Hair among Pregnant Women and Its Effects on Birth Weight and Length in Semarang, Central Java, Indonesia"

_ijerph, 2022, doi:10.3390/ijerph191710684_

Round 1

Reviewer 1 Report

This paper evaluated human MeHg exposure through fish consumption and health effects on children in Semarang, Central Java, Indonesia. The hair Hg levels are not evaluated and the maternal hair Hg level had no association with baby’s birth weight and length. Generally, the study is very common and it did not show significant novelty. I think it can be accepted in this journal after major revision.

 Specific comments:

1. The authors use ppm in whole manuscript. The expression is extremely non-standard. I suggested use mg/kg or µg/g.

2. In the abstract, “Pregnant women living in lowland areas, near the sea, showed higher hair Hg concentration and fish consumption than those in highland areas {[0.465 (0.146 – 8.105) vs 0.385 (0.150 – 1.956) ppm; p = 0.053]”. In table 3, the p value is 0.043. Before and after are inconsistent. Please check it.

3.  In Table 3, for the population who don’t eat fish, the max hair Hg is as high as 8.105 mg/kg. The authors should check it, and I think there are possible inorganic Hg exposure. The hair THg is not suitable for evaluating human MeHg exposure.

Reviewer 2 Report

General:

The paper follows a well-defined methodological framework, and the structure is sufficient. However, the Introduction as well as Discussion should be improved (specific comments are given below). There is also room for improvement within the Results section (specific comments are given below). The results section otherwise nicely follow the aims of the study, however, I miss more in-depth examination of the associations. In particular, analysis of Hg hair levels in relation to different fish species is missing, as the authors stressed in the introduction, that the country-specific advice should be given regarding fish consumption (Lines 82-85) and as described in the Methodology, the info should be available. If this was not possible to analyse, the authors should include this in the Discussion as one of the limitations and explain the reasons and suggestions for future studies.

From the measurements Quality control/Quality assurance perspective, the highest drawback in this study is that the authors did not use the most appropriate reference material to fit the concentration range in hair samples of the study participants better. Is there any specific reason behind this?

Specific comments:

Methyl mercury – as the authors did not perform a speciation study, they should avoid saying that they performed exposure assessment of methylmercury. This should be changed in the title to “assessment of mercury levels in hair”.

Units for Hg concentration used throughout the manuscript: instead of ppm units, the authors should use µg/g, because what they measured is a mass fraction.

Birth outcomes, weight and length, should be specified already in the title, or at least in the abstract under objectives.

Abstract

Nicely and concisely written. The only remark I have here regards the “assessment system” in the sentence: “To establish an assessment system of MeHg exposure through fish consumption and health effects on children, we examined the hair Hg levels in pregnant women and birth outcomes”. What did the authors mean by this? Please clarify or change the term accordingly.

Introduction

Line 40-41: This sentence should be slightly changed: “All forms of Hg have in common a toxic effect, particularly MeHg on the growth and development of the fetus and child”. The emphasis should be on the growth and neurodevelopment as critical effects of exposure to MeHg.

Line 55: Reference no. 13 is a survey protocol (WHO), and does not provide a background for the author’s claim that the assessment of prenatal MeHg exposure is an important issue to be investigated.

Lines 48-78: the review of effects and ‘critical’ values is not well structured. The authors should re-write this part. Maybe start with fish consumption as the main source of exposure, followed by recommendations and then list findings of different epidemiological studies with emphasis on low level exposure. However, I like very much, that the levels of exposure are presented along with the findings of respective studies.

Line 88-98: aims of the study - it should be specified which birth outcomes were used and that MeHg exposure was assessed though total Hg measurements in hair.

Methods

Lines 113-114: I suggest to replace this vague criterion for exclusion of 14-year old mother with the WHO criteria of women in childbearing age (18-40 years).

Lines 118-119: This is true in case primary source of Hg exposure is fish consumption. This should be added.

Lines 121-122: Why was hair cut 1 cm from the scalp and not directly by the scalp? Where were the samples stored before analysis – in a clean room? Why the samples weren’t washed? Did the authors expect negligible external contamination?

Lines 131-135: The author should refine this paragraph. What does the coefficient of variation for Hg measurement of ~4% mean? Is this repeatability among different digests of real samples? The authors report that the measurement was repeated five times and according to reference no. 26, it is stated that this was based on the five repeated measurements using standard hair. This is confusing, because in the last sentence (line 135), the authors mention another CV (1.13%), which was obtained from five times repeated measurements using the CRM. The authors should define the repeatability better.

Line 133: Why only NIES No. 13 was used? This is 10-times the mean level in the study population, therefore also a CRM with lower Hg levels should be used, e.g. IAEA 086. This is a very important aspect of QA/QC.

Line 142: the model of 50 g should be specified better. Was it the same as in the Reference no. 26: the amount consumed per time: 50 g/time or <50 g/time? And how was then the daily amount of consumed fish calculated (< 40, 40-79, 80-159, and >160 g/day)? Was it summed up based on the info for different species? And how can the model of 50 g be useful here? Please define the calculations better.

Lines 154-156: Does this mean, that despite log-transformation, the data was still not normally distributed?

Results

Line 178: is the normal range for the mid-upper arm circumference ≥ 23 cm?

Line 179: I’m not sure whether “no parity” is a correct term? It should be “null parity”. Moreover, the authors refer to 35 women with null parity in the text, while in the table there are 36 women. Please check the numbers.

Lines 184-185: when comparing lowland vs. highland, there should be numbers given for both study groups; e.g. “these were found at a higher frequency in lowland women (n = 14) than in highland women (n = 3) (p = 0.154)”; and clearly stated in the sentence what the authors are comparing, e.g. “Women in lowland areas had a significantly higher income than women in highland areas (p = 0.002)”. Please consider this in all group comparisons.

Line 185-186: The authors did not comment on education level and occupation. At least one sentence should be added.

Lines 193-194: This sentence is redundant.

Table 3: I suggest to include percentile values, e.g. at least p5, p10, p90, p95 along with the median (p50) in the table. This information would be very useful.

Line 233: How did the authors calculate median values for fish consumption (g/day) based on given categories ((< 40, 40-79, 80-159, and >160 g/day) and 50 g model? This is not clear from the description in the methodology. Is it possible that the amount was not correlated significantly to hair Hg because of the high uncertainty in the calculation of the consumption (g/day)? Did you check the association with exclusion of the individual with the highest hair Hg level? This is particularly important, as this subject reported not to consume fish at all… The authors should also compare different categories of fish consumption, as defined in the methodology. What about Hg levels in relation to different species of fish? This would be extremely valuable, to identify which type/species is the most important contributor to Hg exposure in this specific area. Particularly, because this is the first study to deal with exposure to Hg through fish consumption in this area.

Line 234: What history of hair treatment? Do you have more specific info?

Line 254: Small correction in the sentence “Table 4 shows the results of univariate analyses for the association between maternal hair Hg levels with birth outcomes, namely birth weight, and length.”

Lines 255-258: direction of associations should be commented.

Tables 4 and 5: The 95% confidence intervals should be added along with the beta coefficients.

Line 258-260: The authors can not claim there was an association observed, if it is not significant. Should be rewritten in the sense “there was a positive trend between maternal hair Hg and birth weight observed, but the association was not statistically significant”.

Line 261: The authors should add to the Methodology (Statistical analysis) how did they select covariables for the multiple regression model. It seems they did not select them based on univariate analysis?

Discussion

Lines 271-272: The authors cannot claim that this report is about MeHg exposure through fish consumption. Primarily because they did not observe significant association between hair Hg and daily fish consumption.

Lines 279-280: What does it mean “there was no severe complication among their children”. Please define.

Line 283: the authors cannot claim based on their data, that there is no association between Hg exposure and miscarriage, as they only had 2 cases of abortions.

Lines 285-286: the fact that hair samples were not washed before analysis (which would exclude external contamination) should be mentioned here as a limitation of the study.

Line 300: based on what data did the authors claim “In this study, however, most of the subjects did not have these risk factors”? There is no descriptive statistics on alcohol and smoking included in the paper.

Line 325: The authors should also try comparing different categories of fish consumption frequency vs. hair Hg level, and check the association according to different types of fish/seafood. There might be too high uncertainty in g/day for all types of fish…Otherwise, the results show there is another source/determinant of Hg as indicated through hair levels based on which lowland women have higher levels than highland. The only difference in general characteristics between lowland and highland was income. And this should also be mentioned in discussion (please consider using income in the multiple regression model). This might indicate different life style, but the specific characteristics behind this are not known.

Round 2

Reviewer 1 Report

The authors answered all the raised questions. I think it can be accepted at this status.

Author Response

Dear Reviewer,

Thank you for your comment below

"The authors answered all the raised questions. I think it can be accepted at this status"

We submit the English Editing Certificate from Enago. The manuscript was same, but there was a  modification in the article title.

Kind regards,

Authors
